# The Unreasonable Effectiveness of Greedy Algorithms in Multi-Armed Bandit with Many Arms

**Mohsen Bayati**
Stanford University
bayati@stanford.edu

**Nima Hamidi**
Stanford University
hamidi@stanford.edu

**Ramesh Johari**
Stanford University
rjohari@stanford.edu

**Khashayar Khosravi**[*]
Google Research, NYC
khosravi@google.com

## Abstract

We study the structure of regret-minimizing policies in the *many-armed* Bayesian multi-armed bandit problem: in particular, with $k$ the number of arms and $T$ the time horizon, we consider the case where $k \geq \sqrt{T}$. We first show that *subsampling* is a critical step for designing optimal policies. In particular, the standard UCB algorithm leads to sub-optimal regret bounds in the many-armed regime. However, a subsampled UCB (SS-UCB), which samples $\Theta(\sqrt{T})$ arms and executes UCB only on that subset, is rate-optimal. Despite theoretically optimal regret, even SS-UCB performs poorly due to excessive exploration of suboptimal arms. In particular, in numerical experiments SS-UCB performs worse than a simple greedy algorithm (and its subsampled version) that pulls the current empirical best arm at every time period. We show that these insights hold even in a contextual setting, using real-world data. These empirical results suggest a novel form of *free exploration* in the many-armed regime that benefits greedy algorithms. We theoretically study this new source of free exploration and find that it is deeply connected to the distribution of a certain tail event for the prior distribution of arm rewards. This is a fundamentally distinct phenomenon from free exploration as discussed in the recent literature on contextual bandits, where free exploration arises due to variation in contexts. We use this insight to prove that the subsampled greedy algorithm is rate-optimal for Bernoulli bandits when $k > \sqrt{T}$, and achieves sublinear regret with more general distributions. This is a case where theoretical rate optimality does not tell the whole story: when complemented by the empirical observations of our paper, the power of greedy algorithms becomes quite evident. Taken together, from a practical standpoint, our results suggest that in applications it may be preferable to use a variant of the greedy algorithm in the many-armed regime.

## 1   Introduction

We consider the standard stochastic multi-armed bandit (MAB) problem, in which a decision-maker takes actions sequentially over $T$ time periods (the *horizon*). At each time period, the decision-maker chooses one of $k$ arms, and receives an uncertain reward. The goal is to maximize cumulative rewards attained over the horizon. Crucially, in the typical formulation of this problem, the set of arms $k$ is assumed to be "small" relative to the time horizon $T$; in particular, in standard asymptotic analysis

---

[*]The majority of this work was completed while Khashayar Khosravi was affiliated with Stanford University.

of the MAB setting, the horizon $T$ scales to infinity while $k$ remains constant. In practice, however, there are many situations where the number of arms is large relative to the time horizon of interest. For example, drug development typically considers many combinations of basic substances; thus MABs for adaptive drug design inherently involve a large set of arms. Similarly, when MABs are used in recommendation engines for online platforms, the number of choices available to users is enormous: this is the case in e-commerce (many products available); media platforms (many content options); online labor markets (wide variety of jobs or workers available); dating markets (many possible partners); etc.

Formally, we say that an MAB instance is in the *many-armed regime* where $k \geq \sqrt{T}$. In our theoretical results, we show that the threshold $\sqrt{T}$ is in fact the correct point of transition to the many-armed regime, at which behavior of the MAB problem becomes qualitatively different than the regime where $k < \sqrt{T}$. Throughout our paper, we consider a Bayesian framework [2], i.e., where the arms' reward distributions are drawn from a prior.

In §3, we first use straightforward arguments to establish a fundamental lower bound of $\Omega(\sqrt{T})$ on Bayesian regret in the many-armed regime. We note that prior Bayesian lower bounds for the stochastic MAB problem require $k$ to be fixed while $T \to \infty$ (see, e.g., 16, 18, 19), and hence, are not applicable in the many-armed regime.

Our first main insight (see §4) is the importance of *subsampling*. The standard UCB algorithm can perform quite poorly in the many-armed regime, because it over-explores arms: even trying every arm once leads to a regret of $\Omega(k)$. Instead, we show that the $\Omega(\sqrt{T})$ bound is achieved (up to logarithmic factors) by a subsampled upper confidence bound (SS-UCB) algorithm, where we first select $\sqrt{T}$ arms uniformly at random, and then run a standard UCB algorithm [17, 5] with just these arms.

However, numerical investigation reveals interesting behaviors. In Figure 1, we simulate several different algorithms over 400 simulations, for two pairs of $T, k$ in the many-armed regime. [3] Notably, the *greedy* algorithm (Greedy) — i.e., an algorithm that pulls each arm once, and thereafter pulls the empirically best arm for all remaining times – performs extremely well. This is despite the well-known fact that Greedy can suffer linear regret in the standard MAB problem, as it can fixate too early on a suboptimal arm. Observe that in line with our first insight above, subsampling improves the performance of all algorithms, including UCB, Thompson sampling (TS), and Greedy. In particular, the subsampled greedy algorithm (SS-Greedy) outperforms all other algorithms.

The right panel in Figure 1 shows that Greedy and SS-Greedy benefit from a novel form of *free exploration*, that arises due to the availability of a large number of near-optimal arms. This free exploration helps the greedy algorithms to quickly discard sub-optimal arms that are substantially over-explored by algorithms with "active exploration" (i.e., UCB, TS, and their subsampled versions). We emphasize that this source of free exploration is distinct from that observed in recent literature on contextual bandits (see, e.g., 6, 15, 20, 14), where free exploration arises due to diversity in the context distribution. Our extensive simulations in Section 6 and in the longer version of paper [7] show that these insights are robust to varying rewards and prior distributions. Indeed, similar results are obtained with Bernoulli rewards and general beta priors. We refer the interested reader to this longer version of the paper. Further, using simulations, we also observe that the same phenomenon arises in the contextual MAB setting, via simulations with synthetic and real-world data.

Motivated by these observations, in §5 and §7 we embark on a theoretical analysis of Greedy in the many-armed regime to complement our empirical investigation. We show that with high probability, one of the arms on which Greedy concentrates attention is likely to have a high mean reward (as also observed in the right panel of Figure 1). Our proof technique uses the Lundberg inequality to relate the probability of this event to distribution of the ruin event of a random walk, and may be of independent interest in studying the performance of greedy algorithms in other settings. Using this result we show that for Bernoulli rewards, the regret of Greedy is $\tilde{O}(\max(k, T/k))$; in particular, for $k \geq \sqrt{T}$ SS-Greedy is optimal (and for $k = \sqrt{T}$, Greedy is optimal). For more general reward distributions we show that, under a mild condition, an upper bound on the regret of Greedy is $\tilde{O}(\max[k, T/\sqrt{k}])$. Thus theoretically, for general reward distributions, in the many-armed regime Greedy achieves *sublinear*, though not optimal, regret.

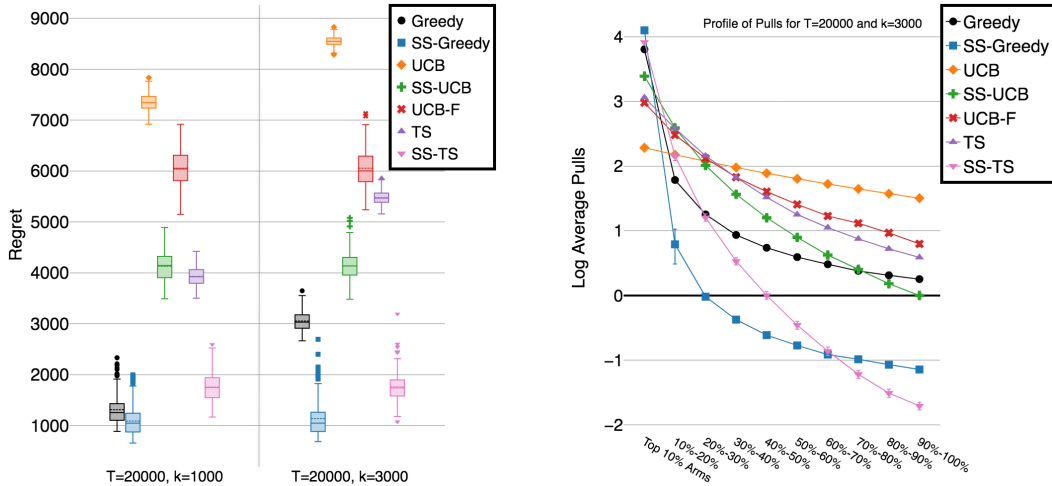

Figure 1: *Distribution of the per-instance regret (on left) and profile of pulls in logarithmic scale based on arms index (on right). Rewards are generated according to $\mathcal{N}(\mu_i, 1)$, with $\mu_i \sim \mathcal{U}[0,1]$. The list of algorithms included is as follows. (1) UCB: Algorithm 1, (2) SS-UCB: Algorithm 2 with $m = \sqrt{T}$, (3) Greedy: Algorithm 3, (4) SS-Greedy: Algorithm 4 with $m = T^{2/3}$ (see Theorem 5), (5) UCB-F: UCB-F algorithm of [27] with the choice of confidence set $\mathcal{E}_t = 2\log(10\log t)$, (6) TS: Thompson Sampling algorithm [26, 22, 2], and (7) SS-TS: subsampled TS with $m = \sqrt{T}$.*

Our theoretical results illuminate why Greedy and SS-Greedy perform well in our numerical experiments, due to the novel form of free exploration we identify in the many-armed regime. Although our theoretical results do not establish universal rate optimality of SS-Greedy, this is clearly a case where regret bounds do not tell the whole story. Indeed, given the robust empirical performance of Greedy and SS-Greedy, from a practical standpoint the combination of our empirical and theoretical insights suggests that in applications it may be preferable to use greedy algorithms in the many-armed regime. This advice is only amplified when one considers that in contextual settings, such algorithms are likely to benefit from free exploration due to context diversity as well (as noted above).

Details of the proofs are all deferred to the longer version of the paper [7].

## 1.1 Related Work

The literature on stochastic MAB problems with a finite number of arms is vast; we refer the reader to recent monographs by [19] and [25] for a thorough overview. Much of this work carries out a frequentist regret analysis. In this line, our work is most closely related to work on the *infinitely many-armed* bandit problem, first studied by [8] for Bernoulli rewards. They provided algorithms with $O(\sqrt{T})$ regret, and established a $\sqrt{2T}$ lower-bound in the Bernoulli setting (a matching upper bound proved by [9]). In [27], the authors studied more general reward distributions and proposed an optimal (up to logarithmic factors) algorithm called UCB-F that is constructed based on the UCB-V algorithm of [4]. In fact, our results in §3 and §4 also leverage ideas from [27]. The analysis of the infinitely many-armed bandit setting was later extended to simple regret [10] and quantile regret minimization [11]. In a related work, [24] proposed using a variant of Thompson Sampling for finding "satisficing" actions in the complex settings where finding the optimal arm is difficult.

Our results complement the existing literature on Bayesian regret analysis of the stochastic MAB. The literature on the Bayesian setting goes back to index policies of [13] that are optimal for the infinite-horizon discounted reward setting. Bayesian bounds for a similar problem like ours, but when $k$ is fixed and $T \to \infty$ were established in [16]; their bounds generalized the earlier results of [18], who obtained similar results under more restrictive assumptions.

Several other papers provide fundamental bounds in the fixed $k$ setting. Bayesian regret bounds for the Thompson Sampling algorithm were provided in [22] and information-theoretic lower bounds on Bayesian regret for fixed $k$ were established in [23]. Finally, [21] proposed to choose policies that

maximize information gain, and provided regret bounds based on the entropy of the optimal action distribution.

## 2 Problem Setting

We consider a Bayesian $k$-armed stochastic bandit setting where a decision-maker sequentially pulls from a set of unknown arms, and aims to maximize the expected cumulative reward generated. In this section we present the technical details of our model and problem setting. Throughout, we use the shorthand that $[n]$ denotes the set of integers $\{1, \ldots, n\}$.

**Time**. Time is discrete, denoted by $t = 1, \ldots, T$; $T$ denotes the time horizon.

**Arms**. At each time $t$, the decision-maker chooses an arm $a_t$ from a set of $k$ arms.

**Rewards**. Each time the decision maker pulls an arm, a random reward is generated. We assume a Bayesian setting, i.e., that arm rewards have distributions with parameters drawn from a common prior. Let $\mathcal{F} = \{P_\mu : \mu \in [0, 1]\}$ be a collection of reward distributions, where each $P_\mu$ has mean $\mu$. Further, let $\Gamma$ be a prior distribution on $[0, 1]$; we assume $\Gamma$ is absolutely continuous w.r.t. Lebesgue measure in $\mathbb{R}$, with density $g$. For example, $\mathcal{F}$ might be the family of all binomial distributions with parameters $\mu \in [0, 1]$, and $\Gamma$ might be the uniform distribution on $[0, 1]$.[4] The following definition adapted from the infinitely-many armed bandit literature (see, e.g. 27, 10) is helpful in our analysis.

**Definition 1** ($\beta$-regular distribution). *Distribution $Q$ defined over $[0, 1]$ is called $\beta$-regular if $\mathbb{P}_Q[\mu > 1 - \epsilon] = \Theta(\epsilon^\beta)$ when $\epsilon \to 0$. Equivalently, there exists $0 < c_0 < C_0$ such that*

$$c_0 \epsilon^\beta \leq \mathbb{P}_Q(\mu > 1 - \epsilon) \leq \mathbb{P}_Q(\mu \geq 1 - \epsilon) \leq C_0 \epsilon^\beta .$$

For simplicity throughout the paper, we assume that $\Gamma$ is 1-regular. [5]

**Assumption 1.** *The distribution $\Gamma$ is 1-regular.*

Assumption 1 is central in our analysis. As shown in §7, our results slightly change for more general $\beta$-regular priors. This definition puts a constraint on $\mathbb{P}[\mu \geq 1 - \epsilon]$, which quantifies how many arms are $\epsilon$-optimal. The larger number of $\epsilon$-optimal arm it is more likely that Greedy concentrates on an $\epsilon$-optimal arm which is one of main components of our theoretical analysis (see Lemma 1). We also assume that the reward distributions are 1-subgaussian as defined below. [6]

**Assumption 2.** *Every $P_\mu \in \mathcal{F}$ is 1-subgaussian: for any $\mu \in [0, 1]$ and any t, if $Z_\mu$ is distributed according to $P_\mu$, then $\mathbb{E}[\exp(t(Z_\mu - \mu))] \leq \exp(t^2/2)$.*

Given a realization $\boldsymbol{\mu} = (\mu_1, \mu_2, \ldots, \mu_k)$, let $Y_{it}$ denote the reward upon pulling arm $i$ at time $t$. Then $Y_{it}$ is distributed according to $P_{\mu_i}$, independent of all other randomness; in particular, $\mathbb{E}[Y_{it}] = \mu_i$. Note that $Y_{a_t,t}$ is the actual reward earned by the decision-maker. As is usual with bandit feedback, we assume the decision-maker only observes $Y_{a_t,t}$, and not $Y_{it}$ for $i \neq a_t$.

**Policy**. Let $H_t = (a_1, Y_{a_1,1}, \ldots, a_{t-1}, Y_{a_{t-1},t-1})$ denote the history up to time $t$ and $\pi$ denote the decision-maker's policy (i.e., algorithm) mapping the history prior to time $t$ to a (possibly randomized) choice of arm $a_t \in [k]$. In particular, $\pi(H_t)$ is a distribution over $[k]$, and $a_t$ is distributed according to $\pi(H_t)$, independently of all other randomness.

**Goal**. Given a horizon of length $T$, a realization of $\boldsymbol{\mu}$, and the realization of actions and rewards, the realized *regret* is then $\mathrm{regret}_T = T \max_{i=1}^k \mu_i - \sum_{t=1}^T Y_{a_t,t}$. We define $R_T$ to be the expectation of the preceding quantity with respect to randomness in the rewards and the actions, given the policy $\pi$ and the mean reward vector $\boldsymbol{\mu}$:

$$R_T(\pi \mid \boldsymbol{\mu}) = T \max_{i=1}^k \mu_i - \sum_{t=1}^T \mathbb{E}[\mu_{a_t} | \pi] .$$

Here the notation $\mathbb{E}[\cdot | \pi]$ is shorthand to indicate that actions are chosen according to the policy $\pi$, as described above; the expectation is over randomness in rewards and in the choices of actions made by

the policy. (In the sequel, the dependence of the preceding quantity on $k$ will be important as well; we make this explicit as necessary.)

The decision-maker's goal is to choose $\pi$ to minimize her Bayesian expected regret, i.e., where the expectation is taken over the prior as well as the randomness in the policy. In other words, the decision-maker chooses $\pi$ to minimize $BR_{T,k}(\pi) = \mathbb{E}[R_T(\pi \mid \boldsymbol{\mu})]$.

**Many arms**. In this work, we are interested in the setting where $k$ and $T$ are comparable. In particular, we focus on the scaling of $BR_{T,k}$ in different regimes for $k$ and $T$.

## 3 Lower Bound

**Theorem 1.** *Consider the model described in §2. Suppose that Assumption 1 holds. Then, there exist absolute constants $c_D$ and $c_L$ such that for any policy $\pi$ and $T, k \geq c_D$, we have*

$$BR_{T,k}(\pi) \geq c_L \min(\sqrt{T}, k).$$

This theorem shows that the Bayesian regret of an optimal algorithm should scale as $\Theta(k)$ when $k < \sqrt{T}$ and as $\Theta(\sqrt{T})$ if $k > \sqrt{T}$. The proof idea is to show that for any policy $\pi$, there is a class of "bad arm orderings" that occur with constant probability for which a regret better than $\min(\sqrt{T}, k)$ is not possible. Interestingly, this theorem does not require any assumption on reward distributions.

## 4 Optimal Algorithms

In this section we describe algorithms that achieve the lower bound of §3, up to logarithmic factors. Recall that we expect to observe two different behaviors depending on whether $k < \sqrt{T}$ or $k > \sqrt{T}$; Theorems 2 and 3 state our result for these two cases, respectively. In particular, Theorem 3 shows that subsampling is a necessary step in the design of optimal algorithms in the many-armed regime. Note that for Theorem 2, instead of Assumption 1, we require the density $g$ to be bounded from above.

We require several definitions. For $i \in [k]$, define: $N_i(t) = \sum_{s=1}^{t} \mathbf{1}(a_s = i)$ and $\hat{\mu}_i(t) = \frac{\sum_{s=1}^{t} Y_{is} \mathbf{1}(a_s = i)}{N_i(t)}$. Thus $N_i(t)$ is the number of times arm $i$ is pulled up to time $t$, and $\hat{\mu}_i(t)$ is the empirical mean reward on arm $i$ up to time $t$. (We arbitrarily define $\hat{\mu}_i(t) = 1$ if $N_i(t) = 0$.) Also define, $f(t) = 1 + t \log^2(t)$.

**Case $k < \sqrt{T}$:** In this case, we show that the UCB algorithm (see, e.g., Chapter 8 of [19]) is optimal (up to logarithmic factors). For completeness, this algorithm is restated as Algorithm 1.

**Theorem 2.** *Consider the setting described in §2. Suppose that Assumption 2 holds and that there exists $D_0$ such that for all $x \in [0, 1], g(x) \leq D_0$. Then, Bayesian regret of Algorithm 1 satisfies*

$$BR_{T,k}(UCB) \leq k[1 + D_0 + D_0(10 + 18 \log f(T))(2 + 2 \log k + \log T)].$$

**Case $k > \sqrt{T}$:** For large $k$, UCB incurs $\Omega(k)$ regret which is not optimal. In this case, we show that the subsampled UCB algorithm (SS-UCB) is optimal (up to logarithmic factors).

**Theorem 3.** *Consider the setting described in §2. Let assumptions 1 and 2 hold. Then, Bayesian regret of the subsampled UCB (Algorithm 2), when executed with $m = \lceil \sqrt{T} \rceil$ satisfies*

$$BR_{T,k}(SS\text{-}UCB) \leq 2 + \sqrt{T} \left[ \frac{\log T}{c_0} + \frac{C_0 \log^2 T}{c_0^2} + C_0(20 + 36 \log f(T))(5 + \log(\sqrt{T} c_0) - \log \log T) \right].$$

---

**Algorithm 1** Asymptotically Optimal UCB

---

1: **for** $t \leq k$ **do**
2:      Pull $a_t = t$
3: **end for**
4: **for** $t \geq k + 1$ **do**
5:      Pull $a_t = \arg\max_i [\hat{\mu}_i(t-1) + \sqrt{\frac{2 \log f(t)}{N_i(t-1)}}]$
6: **end for**

---

---

**Algorithm 2** Subsampled UCB (SS-UCB)

---

1: **Input:** $m$: subsampling size
2: Draw a set of $m$ arms $\mathcal{S}$ uniformly at random (without replacement) from $[k]$
3: Run UCB (Algorithm 1) on arms with indices in set $\mathcal{S}$

---

## 5   A Greedy Algorithm

Motivated by the performance observed in Figure 1 as described in the introduction, in this section we characterize performance of a *greedy* algorithm. The greedy algorithm pulls each arm once and from then starts pulling the arm with the highest estimated mean; the formal definition follows. We can also define a subsampled greedy algorithm that selects $m$ arms and executes greedy on these arms. This is formally defined in Algorithm 4.

---

**Algorithm 3** Greedy

---

1: **for** $t \leq k$ **do**
2:     Pull arm $a_t = t$
3: **end for**
4: **for** $t \geq k + 1$ **do**
5:     Pull arm $a_t = \arg\max_i \hat{\mu}_i(t-1)$
6: **end for**

---

---

**Algorithm 4** Subsampled Greedy (SS-Greedy)

---

1: **Input:** $m$: subsampling size
2: Draw a set of $m$ arms $\mathcal{S}$ uniformly at random (without replacement) from $[k]$
3: Run Greedy (Algorithm 3) on arms with indices in set $\mathcal{S}$

---

**Upper bounds on Bayesian Regret of Greedy.**   We require the following definition. Fix $\theta$. For any $\mu > \theta$, let $\{X_i\}_{i=1}^{\infty}$ be an i.i.d. sequence with $X_i \sim P_\mu$. Let $M_n = \sum_{i=1}^{n} X_i / n$, and define $q_\theta(\mu)$ as the probability that the sample average never crosses $\theta$:

$$q_\theta(\mu) := \mathbb{P}[M_n > \theta \text{ for all } n]. \tag{1}$$

The following lemma provides a general characterization of the Bayesian regret of Greedy, under the weak assumption that rewards are subgaussian (Assumption 2).

**Lemma 1** (**Generic bounds on Bayesian regret of Greedy**). *Let assumption 2 hold. For any* $0 \leq \delta \leq 1/3$, *there holds*

$$BR_{T,k}(\text{Greedy}) \leq T\left(1 - \mathbb{E}_\Gamma\left[\mathbf{1}\left(\mu \geq 1 - \delta\right) q_{1-2\delta}(\mu)\right]\right)^k + 3T\delta$$

$$+ k\mathbb{E}_\Gamma\left[\mathbf{1}\left(\mu < 1 - 3\delta\right) \min\left(1 + \frac{3}{C_1(1 - 2\delta - \mu)}, T(1 - \mu)\right)\right]. \tag{2}$$

*In addition, for SS-Greedy (Algorithm 4) the same upper bound holds, with $k$ being replaced with $m$.*

Lemma 1 is the key technical result in the analysis of Greedy and SS-Greedy. This bound depends on several components, in particular, the choice of $\delta$ and the scaling of $q_{1-2\delta}$. To ensure sublinear regret, $\delta$ should be small, but that increases the first term as $P(\mu \geq 1 - \delta)$ decreases. The scaling of $q_{1-2\delta}$ is also important; in particular, the shape of $q(\cdot)$ will dictate the quality of the upper bound obtained.

Observe that $q_{1-2\delta}(\mu)$ is the only term that depends on the family of reward distributions $\mathcal{F}$. In the remainder of this section, we provide three upper bounds on Bayesian regret of Greedy and SS-Greedy. The first one is designed for Bernoulli rewards; here $q_\theta(\mu)$ has a constant lower bound, leading to optimal regret rates. The second result requires 1-subgaussian rewards (Assumption 2); this leads to a $q$ which is quadratic in $\delta$. The last bound makes an additional (mild) assumption on the reward distribution (covers many well-known rewards, including Gaussians); this leads to a $q$ that is linear in $\delta$, and as a result, a better bound on regret compared to 1-subgaussian rewards.

The bounds that we establish on $q$ rely on Lundberg's inequality, which bounds the ruin probability of random walks and is stated below. For more details on this inequality, see Corollary 3.4 of [3].

**Proposition 1** (Lundberg's Inequality). *Let $X_1, X_2, \ldots$ be a sequence of i.i.d. samples from distribution $Q$. Let $S_n = \sum_{i=1}^{n} X_i$ and $S_0 = 0$. For $u > 0$ define the stopping time $\eta(u) = \inf\{n \geq 0 : S_n > u\}$ and let $\psi(u)$ denote the probability $\psi(u) = \mathbb{P}[\eta(u) < \infty]$. Let $\gamma > 0$ satisfy $\mathbb{E}[\exp(\gamma X_1)] = 1$ and that $S_n \xrightarrow{a.s.} -\infty$ on the set $\{\eta(u) = \infty\}$. Then, $\psi(u) \leq \exp(-\gamma u)$.*

**Bernoulli Rewards.** In this case, we can prove that there exists a constant lower bound on $q$.

**Lemma 2.** *Suppose $P_\mu$ is the Bernoulli distribution with mean $\mu$, and fix $\theta > 2/3$. Then $q_\theta(\mu) \geq \exp(-0.5)/3$, for $\mu \geq (1+\theta)/2$.*

The preceding lemma reveals that for $\delta < 1/6$ and $\mu \geq 1 - \delta$, for the choice $C_{\text{Bern}} = \exp(-0.5)/3$ we have $q_{1-2\delta}(\mu) \geq C_{\text{Bern}}$. We can now state our theorem.

**Theorem 4.** *Consider the model described in §2. Suppose that $P_\mu \sim \mathcal{B}(\mu)$ and the prior distribution satisfies Assumption 1. Then, for $k \geq (30 \log T)/c_0$*

$$BR_{T,k}(Greedy) \leq 1 + \frac{15T \log T}{kc_0} + \frac{3C_0 k}{C_1}\left(5 + \log(kc_0/5) - \log\log T\right).$$

*Furthermore, Bayesian regret of SS-Greedy when executed with $m = \Theta(\sqrt{T})$ is $\tilde{O}(\sqrt{T})$.*

This theorem shows that for $k = \Theta(\sqrt{T})$, Greedy is optimal (up to log factors). Further, for $k \geq \sqrt{T}$, SS-Greedy is optimal.

**Subgaussian Rewards.** In the general case of a 1-subgaussian reward distribution, we prove in longer version of paper [7] that $\inf_{\mu \geq 1-\delta} q_{1-2\delta}(\mu) \geq e^{-1}\delta^2/2$. Combining this and Lemma 1, implies that $BR_{T,k}(\text{Greedy}) = \tilde{O}(Tk^{-1/3} + k)$ and also $BR_{T,k}(\text{SS-Greedy}) = \tilde{O}(T^{3/4})$. While this upper bound on regret is appealing – in particular, it is sublinear regret when $k$ is large — we are motivated by the empirically strong performance of Greedy and SS-Greedy (cf. Figure 1) to see if a stronger upper bound on regret is possible.

**Uniformly Upward-Looking Rewards.** To this end, we make progress by showing that the achieved rate is further improvable for a large family of subgaussian reward distributions, including Gaussian rewards. The following definition describes this family of reward distributions.

**Definition 2.** *Suppose that $Q$ satisfies $\mathbb{E}[Q] = \mu$ and that $Q - \mu$ is 1-subgaussian. Let $\{X_i\}_{i=1}^{\infty}$ be a sequence of i.i.d. random variables distributed according to $Q$ and $S_n = \sum_{i=1}^{n} X_i$. For $\theta < \mu$ define $R_n(\theta) = S_n - n\theta$ and $\tau(\theta) = \inf\{n \geq 1 : R_n(\theta) < 0 \text{ or } R_n(\theta) \geq 1\}$. We call the distribution $Q$ **upward-looking with parameter** $p_0$ if for any $\theta < \mu$ one of the following conditions hold:*

- $\mathbb{P}[R_{\tau(\theta)}(\theta) \geq 1] \geq p_0$

- $\mathbb{E}[(X_1 - \theta)\mathbf{1}(X_1 \geq \theta)] \geq p_0$.

*More generally, a reward family $\mathcal{Q} = \{Q_\mu : \mu \in [0,1]\}$ with $\mathbb{E}[Q_\mu] = \mu$ is called **uniformly upward-looking with parameters** $(p_0, \delta_0)$ if for $\mu \geq 1 - \delta_0$, $Q_\mu$ is upward-looking with parameter $p_0$.*

In the longer version of the paper in [7], we show that a general class of reward families are uniformly upward-looking. In particular, class of reward distributions $\mathcal{F}$ that for all $\mu \geq 1 - \delta_0$ satisfy $\mathbb{E}[(X_\mu - \mu)\mathbf{1}(X_\mu \geq \mu)] \geq c_0$ are $(c_0, \delta_0)$ upward looking. This class includes the Gaussian rewards.

The preceding discussion reveals that many natural families of reward distributions are upward-looking. The following lemma shows that for such distributions, we can sharpen our regret bounds.

**Lemma 3.** *Let $Q$ be upward-looking with parameter $p_0$ which satisfies $\mathbb{E}[Q] = \mu$. Let $\{X_i\}_{i=1}^{\infty} \sim Q$, $S_n = \sum_{i=1}^{n} X_i$ and $M_n = S_n/n$. Then for any $\delta \leq 0.05$, $\mathbb{P}[\exists n : M_n < \mu - \delta] \leq \exp(-p_0\delta/4)$.*

From this lemma, for $\mu \geq 1 - \delta$ we have $q_{1-2\delta}(\mu) \geq 1 - \exp(-p_0\delta/4) \geq (p_0 e^{-1}/4)\delta$. The following theorem shows that in small $\delta$ regime, this *linear* $q$ yields a strictly sharper upper bound on regret than a *quadratic* $q$.

**Theorem 5.** *Let assumptions 1 and 2 hold. Suppose that $\mathcal{F}$ is $(p_0, \delta_0)$ uniformly upward-looking. Then for any $k \geq (4e \log T \max(400, 1/\delta_0^2))/(c_0 p_0)$,*

$$BR_{T,k}(Greedy) \leq 1 + 3T \left[ \frac{4e \log T}{kc_0 p_0} \right]^{1/2} + \frac{3C_0 k}{2C_1} \left( 10 + \log(kc_0 p_0/4e) - \log \log T \right).$$

*Furthermore, Bayesian regret of SS-Greedy when executed with $m = \Theta(T^{2/3})$ is $\tilde{O}(T^{2/3})$.*

It is worth noting that in the case that $k > T$ where subsampling is inevitable, the results presented on SS-Greedy in Theorems 4 and 5 are still valid. The main reason is that our proof technique presented in Lemma 1 bounds the regret with respect to the "best" possible reward of 1 which as stated in Lemma 1 allows for an immediate replacement of $k$ with the subsampling size $m$.

## 6    Simulations

Recall Figure 1 with results of simulations for two pairs of $T, k$ in the many-armed regime where rewards were generated according to Gaussian noise and uniform prior. These results are robust when considering a wide range of beta priors as well as both Gaussian and Bernoulli rewards (see the longer version of the paper [7]). In this section, motivated by real-world applications, we consider a contextual reward setting and show that our theoretical insights carry to the contextual setting as well.

We use the Letter Recognition Dataset [12] from the UCI repository. The dataset is originally designed for the letter classification task (26 classes) and it includes $n = 20000$ samples, each presented with 16 covariates. As we are interested in values of $k > 26$, we only use the covariates from this dataset and create synthetic reward functions as follows. We generate $k = 300$ arms with parameters $\theta_1, \theta_2, \ldots, \theta_k \in \mathbb{R}^d$ ($d$ will be specified shortly) and generate reward of arm $i$ via $Y_{it} = X_t^\top \theta_i + \varepsilon_{it}$. We consider two experiments with $d = 2$ and $d = 6$ and compare the performance of several algorithms in these two cases. As contexts are 16-dimensional, we project them onto $d$-dimensional subspaces using SVD.

For each $d$, we generate 50 different instances, where we pick $T = 8000$ samples at random (from the original 20000 samples) and generate the arm parameters according to the uniform distribution on the $\ell_2$-ball in $\mathbb{R}^d$, i.e., $\theta_i \sim \mathcal{U}_d = \{u \in \mathbb{R}^d : \|u\|_2 \leq 1\}$. We plot the distribution of the per-instance regret in each case, for each algorithm; note the mean of this distribution is (an estimate of) the Bayesian regret. We study the following algorithms and also their subsampled versions (with subsampling $m = \sqrt{T}$; subsampling is denoted by "SS"): (1) Greedy, (2) OFUL Algorithm [1], and (3) TS [26, 22].

**Results.** The results are depicted in Figure 2(a). We can make the following observations. *First*, subsampling is an important concept in the design of low-regret algorithms, and indeed, SS-Greedy outperforms all other algorithms in both settings. *Second*, Greedy performs well compared to OFUL and TS, and it benefits from the same free exploration provided by a large number of arms that we identified in the non-contextual setting: if it drops an arm $a$ due to poor empirical performance, it is likely that it another arm with parameter close to $\theta_a$ is kept active, leading to low regret. *Third*, we find that SS-TS actually performs reasonably well; it has a higher average regret compared to SS-Greedy, but smaller variance. Finally, we see that the performance of Greedy is better for $d = 6$ which highlights that the aforementioned source of free exploration is different from that observed in recent literature on contextual bandits (see, e.g., [6, 15, 20, 14]), where free exploration arises due to diversity in the context distribution. For example, simulations of [6] show that when $d$ is too small compared to $k$, the context diversity is small and performance of Greedy deteriorates which leads to a high variance for its regret (as also seen in the $d = 2$ case here). Figure 2(b) which shows results of the above simulation, but using only $k = 8$ arms, underscores the same phenomena – by reducing the number of arms, the performance of Greedy substantially deteriorates.

## 7    Generalizations and Conclusions

**General $\beta$-regular priors.**   Our results can be extended to $\beta$-regular priors (see Definition 1); see the longer version of paper [7] for more detail. The results are summarized in Table 1. In this table, $\beta_{\mathcal{F}}$ is the exponent of $\delta$ in $\mathbb{E}_\Gamma \left[ \mathbf{1} \left( \mu \geq 1 - \delta \right) q_{1-2\delta}(\mu) \right]$ in Lemma 1. Indeed, it can be shown that

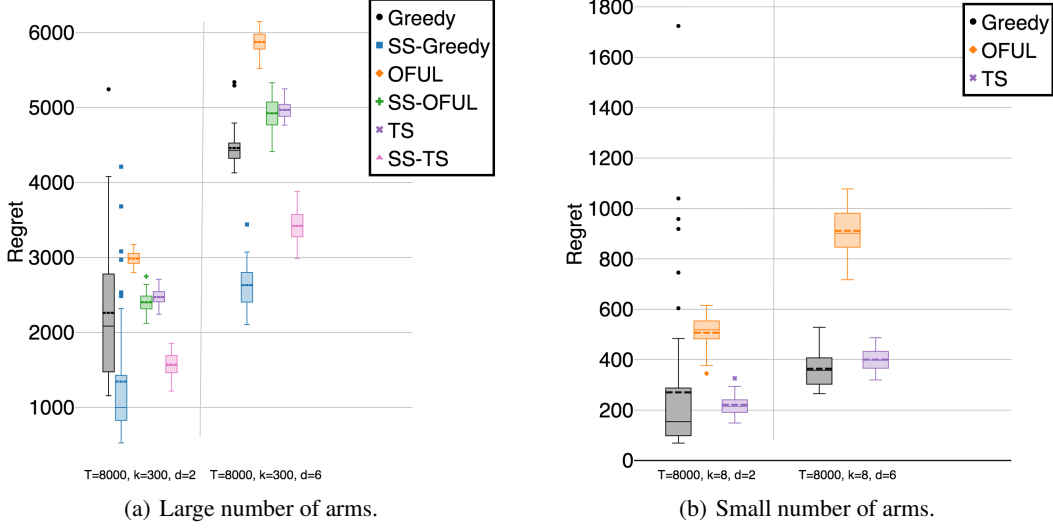

| (a) Large number of arms. | (b) Small number of arms. |
|---|---|

Figure 2: Distribution of the per-instance regret for the contextual setting with real data. For $k = 8$, subsampled algorithms are omitted as subsampling leads to a poor performance. In these figures, the dashed lines indicate the average regret. Comparing the average regrets shows that when covariate diversity is not sufficient, i.e., $d = 2$, the regret of Greedy is much worse for $k = 8$ compared to $k = 300$.

$\beta_{\text{Bernoulli}} = \beta$, $\beta_{\text{Upward-Looking}} = \beta + 1$, $\beta_{\text{1-subgaussian}} = \beta + 2$. Also, SS-UCB and SS-Greedy use all arms whenever $k < m$. For these algorithms the optimal values for $m$ are used.

*Table 1: Regret of various algorithms for different values of $\beta$*

| Algorithm | $\beta < 1$ | | $\beta \geq 1$ | |
|---|---|---|---|---|
| | small $k$ | large $k$ | small $k$ | large $k$ |
| UCB | $k^{1/\beta}$ | $kT^{(1-\beta)/2}$ | $k$ | $k$ |
| SS-UCB | $k^{1/\beta}$ | $\sqrt{T}$ | $k$ | $T^{\beta/(\beta+1)}$ |
| Greedy | $Tk^{-1/\beta_{\mathcal{F}}}$ | $\min(k^{(\beta_{\mathcal{F}}-\beta+1)/\beta_{\mathcal{F}}}, kT^{(1-\beta)/2})$ | $Tk^{-1/\beta_{\mathcal{F}}}$ | $k$ |
| SS-Greedy | $Tk^{-1/\beta_{\mathcal{F}}}$ | $T^{(\beta_{\mathcal{F}}-\beta+1)/(\beta_{\mathcal{F}}-\beta+2)}$ | $Tk^{-1/\beta_{\mathcal{F}}}$ | $T^{\beta_{\mathcal{F}}/(\beta_{\mathcal{F}}+1)}$ |
| Lower Bound | $k$ | $T^{\beta/(\beta+1)}$ | $k$ | $T^{\beta/(\beta+1)}$ |

**Sequential greedy.** When $k$ is large, allocating the first $k$ (or $\sqrt{T}$ arms in case of subsampling) time-periods for exploration before exploiting the good arms may be inefficient. We can design a sequential greedy algorithm, in which an arm is selected and pulled until its sample average drops below $1 - \theta$. Once that happens, a new arm is selected and a similar routine is performed; the pseudo-code of this algorithm is provided in Appendix G. We also show there that for an appropriate choice of $\theta$, Bayesian regret of Seq-Greedy is similar to that of SS-Greedy.

**Future work.** Surprisingly, through both empirical investigation and theoretical development we found that greedy algorithms, and a subsampled greedy algorithm in particular, can outperform many other approaches that depend on active exploration. In this way our paper identifies a novel form of free exploration enjoyed by greedy algorithms, due to the presence of many arms. As noted in the introduction, prior literature has suggested that in contextual settings, greedy algorithms can exhibit low regret as they obtain free exploration from diversity in the contexts. An important direction concerns a unified theoretical analysis of free exploration in the contextual setting with many arms, that provides a complement to the empirical insights we obtain in the preceding section. Such an analysis can serve to illuminate both the performance of Greedy and the relative importance of context diversity and the number of arms in driving free exploration; we leave this for future work.

## Broader Impact

Narrowly, our work is a theoretical study of regret in MABs in the many-armed regime, and as such has no immediate societal consequence. More broadly, however free exploration in general has broader societal consequence, because in many applications, fairness and ethics (and sometimes regulation) preclude active exploration (e.g., healthcare, criminial justice, and education). For this reason, developing an understanding of what is achievable via free exploration is critical to fair and ethical application of MABs in practice.

## Acknowledgements

This work was supported by the Stanford Human-Centered AI Institute, and by the National Science Foundation under grants 1931696, 1839229, and 1554140.

## Footnotes

[2]While our focus is on the Bayesian setting, our analysis can be extended to the frequentist setting.

[3]Our code is available at http://github.com/khashayarkhv/many-armed-bandit.

[4]Our results can be extended to the case where the support of $\Gamma$ is a bounded interval $[a, b]$.

[5]We discuss how our results can be generalized to an arbitrary $\beta$ in Section 7.

[6]Our results generalize to the $S^2$-subgaussian rewards; for brevity we choose $S = 1$ throughout this paper.

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
