[Reviews · NeurIPS 2020]

Review 1

Summary and Contributions: ----------------------------------- After Rebuttal: I have read other reviews and author response. My score remains the same. ---------------------------------- The paper studies the standard multi-armed bandit problems when number of arms is much higher than $O(\sqrt{T})$, $T$ being the time horizon. It first shows that a subsampled version of the UCB algorithm achieves optimal regret in this setting. However, motivated by not-so-good performance of UCB in experiments, the paper studies a simple greedy rule and analyze its performance in the many arm regime.

Strengths: 1. The paper is extremely well-written. The paper flow makes a lot of sense. First, an experimental comparison is shown to motivate the study of the greedy algorithm. Then, a lower bound for the problem is proved and optimal UCB based and greedy algorithms are discussed. 2. The proof of the regret bound of the greedy algorithm incorporates new machinery, which is an welcome addition to the MAB literature.

Weaknesses: The paper considers a week notion of Bayesian regret. It would make the work stronger if similar guarantees for the greedy algorithm can be established in the worst case setting also. I would like the authors to comment on that.

Correctness: I was not able to check the proofs in detail. But the proofs seem correct to me.

Clarity: The paper is very well written and there is a smooth flow of arguments, which helps to understand the main idea pretty well.

Relation to Prior Work: Prior works are well cited and it is clearly mentioned how this work differs from prior works.

Reproducibility: Yes

Additional Feedback:


Review 2

Summary and Contributions: This paper studies the effectiveness of greedy algorithms in the multi-armed bandit with many arms, e.g., k>\sqrt{T}. They show that greedy algorithms with subsampling outperform many other approaches that depend on active exploration. Theoretical analysis and empirical investigation are provided. 

Strengths: This paper first shows the importance of subsampling in the many-armed Bayesian MAB problem with theoretical guarantees. Then simulation results are provided to show the power of greedy algorithms. To explain this phenomenon, the paper gives a theoretical analysis of Greedy in the many-armed regime. Specifically, this paper shows that with high probability, the greedy policy is likely to concentrate on arms with high mean rewards. And the paper derives an upper bound on regret of SS-Greedy for Bernoulli rewards and more general reward distributions under a mild condition.

Weaknesses: As mentioned in the paper, SS-Greedy does not establish universal rate optimality, although it shows better empirical performance in simulations provided in the paper. Experimental results for subgaussian rewards and uniformly upward-looking rewards explored in the theoretical parts are not shown. It seems that this paper does not show whether their analysis holds when k > T, where is realistic, as discussed at the beginning of the paper. ****************** Thanks for your response. I will hold my current score because my concerns are not fully addressed. For example, the sentence for the motivating example ‘In practice, however, there are many situations where the number of arms is large relative to the time horizon of interest’ In Section 1 brings the case with k>T to me. It is very unclear for me that the current algorithm will work for the case with k>T.

Correctness: Seems correct.

Clarity: Satisfied.

Relation to Prior Work: The relations of theoretical results and related work are not clear. 

Reproducibility: Yes

Additional Feedback: A summary for contributions in the first section should be provided Some notations, e.g., free exploration and subsampling, should be explained/defined before they are used. Typos: Line 54: behavior->behaviors Line 69: reward -> rewards Line 68: in supplementary -> in the supplementary Line 76: Figure 1 ) -> Figure 1) The tense in Section 1.1 Related Work should be consistent. References, e.g., [21], should not be used as a noun.  Line 267: Figure 6 -> Figure 2


Review 3

Summary and Contributions: This paper continues along the interesting recent direction of characterizing the performance of a simple greedy algorithm in bandit optimization. The main problem considered is the many-armed Bayesian MAB problem, i.e., where the number of arms k > sqrt{T}. Motivated by the empirical findings in this regime, the authors study the theoretical performance of popular MAB algorithms and provide insights into the effectiveness of greedy algorithms. The main contributions include: 1) lower bound on the Bayesian expected regret, 2) two algorithms based on the simple idea of subsampling: SS-UCB and SS-Greedy, 3) optimal regret rates for SS-UCB and (sub)optimal regret rates for SS-Greedy for different types of rewards.

Strengths: The problem considered in this paper is very well motivated by the interesting empirical study presented in the introduction. Recently, there has also been an increasing interest in the line of works that study the performance of the greedy algorithm in bandit optimization (e.g., in the contextual setting). This work seems to be the first to study the performance (in the Bayesian setting) of both the standard bandit and greedy algorithms in the regime of many arms. The obtained theoretical results in the many arms regime seem novel. These are of interest since in many real-world problems the number of arms can be huge, while the number of optimization rounds can be limited. The obtained regret rates are supported by the lower bound performance. A simple idea of subsampling seems to work well in both theory and practice. The authors go one step further to show that the better regret rates can be obtained for a special (but still a large) family of sub-gaussian reward distributions. Some non-standard tools are used in the analysis such as Lundberg inequality. There are also some interesting additional contributions in Section 7, i.e., going beyond 1-regular priors.

Weaknesses: Overall, I do not find any very major weaknesses when it comes to this paper. Apart from that, it seems that Assumption 1 is crucial in obtaining results of almost every theorem. However, although the statement of this assumption is easy to understand, I feel that the role (intuition, importance) of this assumption in the theoretical analysis is not properly discussed. Next, the main algorithmic idea is very simple, and it is not clear whether the obtained upper bounds for the (ss)-greedy algorithms and sub-gaussian rewards can be improved. Currently, there is a linear dependence on T in the small k regime. Is this unavoidable? Finally, I have some concerns when it comes to Section 6. Please see below for more details.

Correctness: I haven’t checked the proofs in the appendix carefully, but I did not find a good reason to believe that the presented results are incorrect.

Clarity: Overall, the paper is clearly written, however, the paper would benefit from more intuition/explanations when it comes to the main used assumptions.

Relation to Prior Work: This paper clearly discusses the connection to previous works and how it differs from other recent ones that also study the performance of greedy algorithms in some related but different settings.

Reproducibility: Yes

Additional Feedback: Post-rebuttal comments: I've read the rebuttal and other reviews. The authors have addressed most of my concerns and hence I increase my score. I hope the authors would make the suggested edits in the revised version and explain the role of their main assumption. _______________________________________________ Some comments/questions: -- Assumption 1 seems to be the main ingredient of all the main theorems, while the intuition of this assumption and its impact on the analysis (i.e., challenges) are not properly discussed. Can you explain why things fail if this assumption does not hold? -- Why is Assumption made in Theorem 2 stronger compared to Assumption 1? How are these compared? -- What are the inherent difficulties in the analysis of the greedy algorithm when it comes to sub-Gaussian rewards (in comparison to Bernoulli rewards) that result in higher regret rates? -- In the case of sub-gaussian rewards, the bounds for (ss)greedy do not match the lower bounds, and in the small k regime bounds are linear in T. Currently, it is not clear if these bounds can be improved. -- Are there other ideas for subsampling other than uniform sampling? Can you make use of a prior (in the case it is informative)? -- Can you elaborate on considering the Bayesian framework for the “many arms” problem? Can similar insights/results be obtained in the frequentist setting? Simulations: -- Can you comment on the high variance of the performance of greedy (when d=2) in Figure 2 in comparison to its performance when d=6? Next, why is the variance of the greedy methods larger than in the case of TS or OFUL? This raises the question of whether we should really care about the expected performance when it comes to the greedy algorithm. -- Why would contextual simulations be considered since this is not the actual setup studied in the paper? In particular, the context diversity can be present here as well, and for which we know that the greedy algorithm works well. Can the results be due to this form of exploration only?! -- “Performance of Greedy is better for d=6, due to another source of exploration…” -- What is this conclusion based upon?

[Author Response · NeurIPS 2020]

We thank the review team for the extensive set of comments and suggestions. Here, we detail our responses to your
comments. Due to space limitations, the reviewers' comments are summarized in *italic*. We also adopt the same
numbers for the references as the submission.

**Reviewer 2.** *Bayesian vs frequentist regret.* This is a great point. We actually establish our Bayesian regret bound by
first deriving frequentist upper bounds on regret and then taking expectation with respect to the prior (see lines 532-537
on page 17). We used Bayesian notion throughout the paper which allows for an easier exposition of the ideas/results.
We would be happy to add a discussion on the Bayesian vs frequentist regret in our final version.

**Reviewer 3.**

• *SS-Greedy does not establish universal rate optimality.* This is correct, but as mentioned in the abstract, this is one
of the cases that the universal optimality does not tell the whole story and our empirical observations suggest that
SS-Greedy performs very well in practice. We leave improved analysis of regret rates for SS-Greedy for future work.
• *Experimental results for subgaussian and uniformly upward-looking not included.* Due to space limitations, we
pushed most of our simulations to the appendices. In fact, we include simulations over a wide range of beta priors
with both Bernoulli and Gaussian reward (uniformly upward-looking as shown in Appendix C) in Appendix E.
• *Extension of results to $k > T$.* Thanks for bringing this to our attention. All our results can be extended to the case
that $k > T$ (here the sub-sampling is inevitable). We will add a remark on this point in our final version.
• *Theoretical results and comparison with prior work.* Due to space limitations, we decided to discuss the most related
works and refer the reader to the recent monographs by [18] and [23]. However, for improving the presentation of the
paper, we will expand the related work section to address your point.
• *Summary of contributions, notations and typos.* Thanks for this suggestion and pointing out the typos. We will add a
summary of contributions, fix all the typos and define the notations that have not been defined in the final version.

**Reviewer 4.**

• *Assumption 1 and its role in theoretical analysis.* You are absolutely right that Assumption 1 is central in our analysis.
As shown in Section 7, our results slightly change for more general $\beta$-regular priors. This definition puts a constraint
on $\mathbb{P}[\mu > 1 - \epsilon]$, which quantifies how many arms are $\epsilon$-optimal. The larger number of $\epsilon$-optimal arm means it
is more likely that Greedy concentrates on an $\epsilon$-optimal arm which is one of main components of our theoretical
analysis. It would be interesting to remove all the assumptions on the prior, but it seems that we should not hope to
get any result better than the well-known worst-case regret of $\Omega(\sqrt{kT})$ for arbitrary priors as all mass can be put on
difficult problems (see Section 35.1 of [18]). Alternatively, one can replace it with some other assumption, however,
we decided to use $\beta$-regular priors adapted from the literature on infinitely many-armed bandits [10, 25].
• *Simplicity of algorithmic idea and regret improvement.* A key (and practical) benefit that the Greedy algorithm offers
is its simplicity. In our analysis, we observed that most subgaussian rewards are uniformly upward-looking and that
is the main reason that we proved tighter regret bounds for this family. It is an interesting future direction to improve
the regret bounds for subgaussian rewards. The linear regret for small values of $k$ is an inherent property of greedy
algorithms as the lack of active exploration leads to a linear regret with some probability. We prove that if $k$ is large,
this probability is small (and discuss how it decays depending on $k$ for different rewards) and hence its contribution
to total regret is negligible.
• *Comparison of assumption in Theorem 2 and Assumption 1.* Thanks for bringing this to our attention. In the small $k$
regime, the lower bound on $\mathbb{P}[\mu > 1 - \epsilon]$ in Assumption 1 is not needed for establishing regret bounds. Hence, the
assumption that density $g(x) \leq D_0$ for all $x \in [0, 1]$ implies that $\mathbb{P}[\mu \geq 1 - \epsilon] \leq D_0\epsilon$, implying the upper bound in
Assumption 1. We will revise this sentence in the final version to address this.
• *Difficulties in analysis of Greedy & discussion of regret for different families.* The main difficulty in analyzing Greedy
algorithms is that it is possible that all the good arms get poor observations at the beginning and hence the algorithm
would stick to a sub-optimal arm, leading to a linear regret. We can be hopeful that the probability of this event is
small if number of arms is large and is indeed a key component in our analysis of the Greedy algorithm. In particular,
the quantity $q_\theta(\mu)$ (defined in Eq. (1)) captures how likely it is for a good arm to have its sample mean drop below $\theta$
(or get poor observations). For the exact same reason, the shape of $q_\theta(\mu)$ dictates the final regret bound and is the
main reason for getting different rates for Bernoulli, subgaussian, and uniformly upward-looking distributions.
• *Informative priors.* This is an interesting point. If all arms have the same prior, the Bayesian regret should not depend
on sampling strategy. But you are right that for different arm priors, uniform sampling can be sub-optimal and this
would be an interesting future direction.
• *Comments on simulations.* Our intent in including simulations with both $d = 2$ and $d = 6$ is precisely to shed light
on the relative impact of context diversity versus the presence of many arms, on the (good) performance of the greedy
algorithm. Informally, we expect that for $k = 300$ and $d = 2$, context diversity is insufficient to ensure adequate free
exploration; indeed, while greedy performs reasonably on average, the variance is high – the opposite of what we
would expect if context diversity were sufficient (see, e.g., [6]). On the other hand, for $d = 6$, we can likely attribute
the lower variance in this case to the context diversity, providing an additional source of free exploration that also
contributes to the better performance of greedy. As we mainly consider Bayesian regret in our paper, we compare the
expected performance, but we agree that in some applications it would be beneficial to consider other metrics.

[Meta-Review · NeurIPS 2020]

All reviewers agree that the paper considers a problem of relevance (bandits with many arms) and shows interesting results about simple-to-implement learning algorithms based on the greedy principle. However, one lingering concern that arose during the discussions among the reviewers was whether/how the results obtained in the paper applied for the case when the number of arms is larger than the time horizon of the game (k >T). It appears that the author response to this question has not been substantial. Though I can see that this will not be an issue -- the proof of Lemma 2 bounds regret with respect to the best possible reward of 1, the author(s) is/are requested to add a precise clarification of this regime in the updated version.